# High Ferritin Is Not Needed in Hemodialysis Patients: A Retrospective Study of Total Body Iron and Oral Iron Replacement Therapy

**DOI:** 10.3390/ijms25031508

**Published:** 2024-01-25

**Authors:** Chie Ogawa, Ken Tsuchiya, Naohisa Tomosugi, Kunimi Maeda

**Affiliations:** 1Maeda Institute of Renal Research, 6F-1-403 Kosugi-cho, Nakahara-ku, Kawasaki 211-0063, Kanagawa, Japan; kuni@maeda-irr.com; 2Biomarker Society, INC, 6F-1-403 Kosugi-cho, Nakahara-ku, Kawasaki 211-0063, Kanagawa, Japan; tsuchiya@twmu.ac.jp (K.T.); tomosugi@kanazawa-med.ac.jp (N.T.); 3Department of Blood Purification, Tokyo Women’s Medical University, 8-1 Kawada-cho, Shinjuku-ku, Tokyo 162-8666, Tokyo, Japan; 4Division of Systems Bioscience for Drug Discovery Project Research Center, Medical Research Institute, Kanazawa Medical University, 1-1 Daigaku, Kahoku-gun, Uchinada-machi 920-0293, Ishikawa, Japan

**Keywords:** total body iron, serum ferritin, body surface area, hemodialysis, anemia, oral iron replacement therapy

## Abstract

In vivo iron levels can be adjusted through intestinal iron absorption to be maintained at a suitable level; however, optimal iron levels in hemodialysis (HD) patients are unclear. In this study, we investigated total body iron (TBI), calculated as the sum of red blood cell (RBC) iron and iron stores, during courses of low-dose oral iron replacement therapy, and evaluated in vivo iron sufficiency and its indicators in HD patients. We analyzed data on 105 courses of low-dose iron replacement therapy administered to 83 patients on maintenance HD over 7 months. We evaluated changes in TBI, RBC iron, and iron stores from the initiation of treatment to month 7 in two groups of patients, namely, iron-therapy responders and non-responders. TBI showed significant increases until month 4 and plateaued thereafter in iron-therapy responders, and tended to increase and then reached a similar plateau in non-responders (month 7: 1900 ± 447 vs. 1900 ± 408 mg). Steady-state TBI was strongly correlated with body surface area (y = 1628.6x − 791.91, R^2^ = 0.88, *p* < 0.001). We observed constant TBI during oral iron replacement therapy suggesting the activation of a “mucosal block”. The results suggest that body surface area has utility for estimating the required TBI with regression equations.

## 1. Introduction

Iron is a component of hemoglobin (Hb) and plays a vital role in the differentiation and proliferation of immature red blood cells (RBCs) [1,2], making it indispensable for hematopoiesis. Hb is synthesized during the process of hematopoiesis, which is promoted through receptor binding of the erythropoietin released in response to hypoxia. Iron deficiency actively inhibits the degradation of hypoxia-inducible factor alpha (HIF-α) because the prolyl hydroxylase domain enzymes, which catalyze HIF-α degradation, are iron dependent [3]. Iron is thus closely associated with hematopoiesis and hypoxia-inducible factors, and its retention is vital for the living body.

The in vivo regulation of iron involves its absorption from the intestines and its maintenance at appropriate levels, which is important because excess iron can lead to oxidative stress. This mechanism has been referred to as the “mucosal block”. The body’s normal iron content is 3–4 g and is held mainly in the following two ways: in the reticuloendothelial system (approximately 70%) or as ferritin in the liver (approximately 30%). The daily amount of iron needed for hematopoiesis is around 20–25 mg, which comes predominantly from recirculated iron. Iron has no active excretion pathway from the body and circulates in a semi-closed system; therefore, hematopoiesis depends on satisfactory iron metabolism. The main regulator of iron metabolism is hepcidin. Iron supply to the blood is mainly provided by intestinal cells, reticuloendothelial macrophages, and hepatocytes via FPN. When hepcidin binds to ferroportin, the conjugate is degraded in lysosomes and iron cannot be transported from the cell to the bloodstream. Hepcidin is upregulated by iron signaling or inflammation and negatively regulates iron metabolism [4]. Iron deficiency is classified into absolute iron deficiency, in which the total body iron is deficient, and functional iron deficiency, in which iron deficiency in the blood occurs due to elevated hepcidin despite sufficient total body iron.

Renal anemia in HD patients results from decreased EPO production due to the deficiency of HIF2α. Therefore, therapeutic approaches to renal anemia in hemodialysis (HD) patients center on HIF–prolyl hydroxylase domain inhibitors, which have recently become commercially available, as well as erythropoietin-stimulating agents (ESAs). However, HD therapy is still associated with iron loss [5,6]. Accordingly, iron replacement therapy is also regarded as an important approach to managing anemia in HD patients. However, the optimal iron level in HD patients experiencing chronic inflammation is unclear because serum ferritin, a major marker of iron, is affected by inflammation. Furthermore, inflammation-induced hyperhepcidinemia can lead to iron deficiency due to the malabsorption of iron from the intestinal tract; in Europe and the United States, the recommended treatment approach involves non-physiological intravenous iron replacement [7,8]. However, it has been reported that iron metabolism may not significantly differ between HD patients and healthy individuals [9,10]. Meanwhile, The Dialysis Outcomes and Practice Patterns Study (DOPPS) data show that Japanese HD patients have lower ferritin levels and better inflammation control [11,12], and oral iron replacement therapy is reported to be effective [13,14,15]. Iron overload may elevate hepcidin levels, which in turn may cause impaired iron utilization and oxidative stress. It is important to clarify the optimal amount of iron in HD patients.

Serum ferritin is useful as an indicator of iron stores, and normally it can be used to determine any excess or deficiency of iron in the body of healthy individuals with stable hematopoiesis [16]. However, the fact that iron circulation is a semi-closed system has some implications. RBC iron and iron stores function together as the two major in vivo reservoirs of iron, and serum ferritin does not necessarily reflect body iron levels in individuals with unstable hematopoiesis.

To address these issues, researchers have used total body iron (TBI) as an index of in vivo iron levels. TBI is the sum of the iron stores, while RBC iron calculated from Hb, serum ferritin, and soluble transferrin receptor levels. When patients receive iron replacement therapy to treat iron-deficiency anemia, they first experience an improvement in the Hb level, followed by a replenishment of iron stores. Estimating the in vivo iron level using Hb or serum ferritin alone would be challenging, but both parameters are reflected in the TBI value, making it suitable for ascertaining the approximate amount of iron in vivo. An increase in the TBI level is considered to reflect the absorption of iron during oral iron replacement therapy, and TBI has been used in a number of studies [17,18,19,20]. Cable et al. investigated iron kinetics in blood donors and demonstrated the efficacy of orally administered iron [17], while Bialkowski et al. reported similar efficacy in patients receiving iron orally at 19 mg or 38 mg [18]. Furthermore, Cable et al. were able to calculate iron stores by using only the serum ferritin level, which are collected as part of patients’ regular care, without using soluble transferrin receptor levels [17].

Against this background, in the present study, we investigated dynamic changes in TBI as well as erythrocyte/iron-related parameters in HD patients on physiological oral iron replacement therapy in order to determine TBI sufficiency values, and to evaluate markers indicative of TBI sufficiency.

## 2. Results

### 2.1. Patients

During the observation period, iron supplementation was administered intravenously to 17 patients due to gastrointestinal disorders, and 124 patients received oral iron supplementation.

The target population consisted of 88 HD patients who had undergone a total of 111 courses of iron therapy over seven months, excluding patients with hemorrhage and increased ferric citrate hydrate dose due to high phosphorus levels. After excluding five patients whose measured Hb levels were consistently below 12 g/dL (6 times), eighty three patients who had undergone a total of 105 courses of iron replacement therapy were analyzed. For patients who had undergone multiple courses of iron replacement therapy, their age and time on hemodialysis were recorded when blood sampling was performed at the start of each course (month 0). Mean age was 68.1 ± 12.6 years and median time on hemodialysis was 6.6 years. Of the 105 courses of therapy, 75 were delivered to men and 30 to women, and 66.7% of the patients had diabetic nephropathy. The patients had a mean Hb level of 10.4 ± 0.7 g/dL, a mean serum ferritin level of 27.3 ± 12.5 ng/mL, mean transferrin saturation (TSAT) of 18.1 ± 5.9%, and a median C-reactive protein level of 0.1 mg/dL. Patients received iron as ferrous citrate in 64 cases (50 mg/day, *n* = 56; 100 mg/day, *n* = 8) and as ferric citrate in 41 cases (250 mg/day, *n* = 39; 500 mg/day, *n* = 2) (Table 1).

### 2.2. Changes in TBI, RBC Iron, and Iron Stores

Regarding TBI, iron-therapy responders showed significant monthly increases up to month 4 and maintained a constant value thereafter, whereas non-responders showed a tendency toward a gradual increase, without any significant monthly differences up to month 4, and maintained a constant value thereafter, similar to that for the responders (Figure 1a). As for RBC iron, iron-therapy responders showed significant monthly in-creases up to month 3, whereas non-responders showed constant values (Figure 1b). Regarding iron stores, iron-therapy responders showed significant monthly increases up to month 5, while non-responders showed a significant increase only at month 1 (Figure 1c). The chronological changes in TBI, RBC iron, and iron stores differed significantly between iron-therapy responders and non-responders (*p* < 0.001, Figure 1a–c), although neither group showed significant differences in the pre-treatment data (Table 2).

On the other hand, serum ferritin showed a significant increase in both groups at month 1, but thereafter a significant increase was only observed at month 5 in the responder group, and the change over time was not different between the two groups (*p* = 0.72, Figure 1d). DA per body weight showed a significant decrease at month 3 and 4 in the effective group, but a slight increase in the non-responder group, and the change over time was also significantly different between the two groups (*p* < 0.001, Figure 1e).

### 2.3. Correlates with Steady-State TBI

Among the 105 courses of iron replacement therapy, TBI achieved a steady state in 104 courses and was persistently elevated in only 1 course. Steady-state TBI did not correlate with Hb or serum ferritin but was strongly correlated with body surface area (BSA) [TBI = −791.914 + 1628.606 × BSA (m^2^); R^2^ = 0.88, *p* < 0.001, Figure 2]. The correlation between steady-state TBI and BSA did not differ between iron-therapy responders and non-responders (*p* = 0.55; Figure 3).

### 2.4. Steady-State TBI and Erythrocyte/Iron-Related Parameters

Hb levels were significantly higher in iron-therapy responders than in non-responders (11.3 ± 0.5 vs. 10.8 ± 0.6 g/dL, *p* < 0.001; Figure 4a). Serum ferritin levels did not differ between iron-therapy responders and non-responders (Figure 4b). TSAT was significantly higher in iron-therapy responders than in non-responders (28.7 ± 10.0% vs. 20.8 ± 7.7%, *p* < 0.001; Figure 4c).

### 2.5. Correlation between BSA-Adjusted TBI and Hepcidin

In the sensitivity analysis, the differences between the slopes above and below the hepcidin cut-off points were evaluated with the F-statistic for interaction. The cut-off point showing the greatest difference (inflection) was determined to be the optimal cut-off point, which in this study was 15 ng/mL (F = 38.74, Table 3). A significant positive correlation was lost at cutoff points above 25 ng/mL, with no further augmentation of BSA-adjusted TBI in line with any increase in hepcidin (Figure 5).

## 3. Discussion

Renal anemia is an almost inevitable complication of CKD patients and is defined as anemia resulting from the impaired production of erythropoietin (EPO) in the kidney; a deficiency in EPO reduces the hematopoietic stimulation of the bone marrow, resulting in anemia. As for molecular biological mechanisms, the EPO protein was purified, EPO-producing cells in the kidney were identified in 2008 [21], and a series of studies on hypoxia-inducible factors revealed that the reduced production of EPO is induced by weakened HIF2α [22,23]. On the other hand, in the past, its treatment required frequent blood transfusions, leading to iron accumulation and the risk of infectious diseases such as hepatitis, which particularly impaired the quality of life of dialysis patients. However, in 1990, the clinical application of synthetic erythropoietin (recombinant human erythropoietin: rHuEPO), now collectively called ESA (erythropoietin stimulating agent), became possible, and especially the clinical application of long-acting ESA has had a therapeutic effect. Furthermore, since 2019, HIF-PH inhibitors have been available for clinical use [24]. Thus, the treatment of renal anemia, along with the application of various preparations, now involves the establishment of optimal Hb levels and, in particular, information on iron supply, which is essential for efficient hematopoiesis. Progress has also been made in identifying the regulatory mechanisms of iron metabolism and related proteins, and understanding iron kinetics and adequate iron supply, along with ESA or HIF-PHI administration, are essential prerequisites for stable renal anemia management.

Our previous study investigating reticulocyte hemoglobin content and hepcidin suggested that iron deficiency is possible at serum ferritin levels below 60 ng/mL [25]. Accordingly, we obtained data from patients receiving small-dose iron replacement therapy to address serum ferritin levels of less than 60 ng/mL and determined their TBI and evaluated changes in this parameter. In HD patients, changes in TBI are considered to reflect physiological iron absorption because small-dose oral replacement therapy is dependent on physiological iron absorption.

Oral iron supplementation is considered to be ineffective in HD patients because they tend to have chronic inflammation, which severely impacts their ability to absorb iron [7,8]. Accordingly, we divided the patients for evaluation in this study into iron-therapy responders and non-responders in order to investigate whether iron absorption differs according to the effectiveness (or lack thereof) of oral iron supplementation.

In this study, there was a significant increase in TBI after the initiation of iron replacement therapy, which then reached a plateau in iron-therapy responders. These findings suggest that a mucosal block is activated when small-dose iron replacement therapy is continued after iron sufficiency is achieved, and that TBI will not continue to increase thereafter. In iron-therapy non-responders, TBI tended toward a gradual, non-significant increase in the early phase after the initiation of treatment, and then plateaued from month 4 onwards in the same way as in responders. Although non-responders showed a smaller increase in TBI compared with responders, they were considered to have physically absorbed the necessary amount of iron, based on evaluations of the change in TBI and steady-state TBI. However, non-responders showed no increase in RBC iron, and their Hb levels upon achieving steady-state TBI were also significantly lower than those in responders. Furthermore, TSAT was significantly lower in iron-therapy non-responders, although the serum ferritin level did not differ between responders and non-responders. According to reports from the Japanese Society for Dialysis Therapy, Hb is decreased at a TSAT of less than 20% and the TSAT influences the effectiveness of ESAs in HD patients [26]. We suggest that differences in the amount of utilizable iron for hematopoiesis, rather than intestinal iron absorption, may account for any differences in the effect of oral iron replacement therapy.

Hepcidin regulates both intestinal iron absorption and the reticuloendothelial-macrophage iron supply to the blood [4]. However, the responses it elicits in reticuloendothelial macrophages and intestinal cells are reported to differ; the former shows a sensitive response whereas the latter does not [27]. Since ESAs can also transiently lower hepcidin [28,29], intestinal iron absorption may be possible in patients with mildly elevated hepcidin. Furthermore, independently of hepcidin, hypoxia-inducible factors are reported to induce the expression of cytochrome b and DMT-1 as well as promote iron absorption from the intestinal tract [30,31]. Thus, intestinal iron absorption may not necessarily match the supply of iron for utilization in hematopoiesis.

We evaluated the relationships between steady-state TBI and independent factors in simple regression analyses in order to determine whether any factor was indicative of TBI achieving a steady state. The strongest correlation we found was between steady-state TBI and BSA, for which the regression equation yielded a high coefficient of 0.88, suggesting that BSA has utility for calculating TBI sufficiency. However, steady-state TBI did not appear to correlate with serum ferritin, suggesting that it might not be reliable as a marker of TBI sufficiency. Serum ferritin was below 100 ng/mL in many patients with sufficient TBI, indicating that in vivo iron levels may be adequate even at serum ferritin levels lower than the threshold values for iron supplementation stated in the guidelines. TBI levels of 1200–2700 mg have been found to be sufficient for Japanese HD patients of slight stature with a target Hb of 10–12 g/dL, and such TBI levels are lower than those commonly regarded as acceptable.

This was a retrospective study analyzing data acquired from patients undergoing regular medical care, and thus we did not measure hepcidin, which is a regulator of iron homeostasis. However, we did evaluate the correlations between hepcidin and TBI (adjusted for BSA), using data obtained in September 2008 from patients provided darbepoetin alfa (DA) in the same way as in the present study to manage anemia. This study showed that the required TBI is strongly dependent on BSA. Therefore, the BSA-adjusted TBI is expected to represent the degree of TBI sufficiency. If TBI is sufficient, TBI is presumed to remain constant with hepcidin-induced mucosal block. The results of the present investigation indicated that iron absorption occurred at a hepcidin level of 15 ng/mL or below, and a possible mucosal block activation occurred at hepcidin levels exceeding 15 ng/mL.

The results of our previous study indicated that TSAT is vital for Hb synthesis and suggested the hepcidin levels that may be optimal for that process [25]. Large-dose iron supplementation is known to result in elevated hepcidin [32], and inappropriate iron administration may lead to a reduction in available iron (i.e., functional iron deficiency) despite iron sufficiency in vivo. When iron supplementation is provided, it is important to select a method of administration that does not elevate hepcidin.

Recently, the PIVOTAL study reported that high-dose intravenous iron administration is effective in improving prognosis and reducing the dose of ESA [33]. In addition, intravenous formulations of high-titer iron are now available on the market, allowing large amounts of iron to be administered intravenously in a single dose. Ferric carboxymaltose and ferric derisomaltose, which can be administered intravenously in large doses at once resulting in a reduction in oxidative stress, has been reported to improve anemia, reduce ESA, and increase ferritin and TSAT [34,35,36,37,38,39,40]. However, it has been suggested that high-dose iron administration and a high ferritin level are associated with iron deposition in the liver [41,42] and iron deposition in the liver and heart has also been reported to be observed in many hemodialysis patients [43]. Therefore, although high-dose intravenous iron administration may be useful in the short term, concerns remain about its long-term safety. In this study, the average serum ferritin and TSAT in the steady state of responders group were 78.6 ng/mL and 28.7%, respectively, and high TSAT was maintained even with low serum ferritin, indicating good iron utilization efficiency. It has been shown that in such patients, it is possible to improve anemia and reduce ESA dosage even with low doses of oral iron supplements and a low ferritin level.

From the present findings, we consider that it is difficult to determine iron sufficiency based on serum ferritin alone and that prediction of steady-state TBI using BSA is useful for avoiding excessive iron administration. However, TBI is determined based on serum ferritin, and thus satisfactory management of inflammation may be necessary in order to make effective use of TBI.

This was a single-center study with a small analysis set comprising patients who were receiving regular medical care. In the future, it will be necessary to conduct multi-center, prospective studies in larger populations.

## 4. Materials and Methods

### 4.1. Patients

We targeted 88 patients undergoing maintenance hemodialysis at our institution, for whom small-dose oral iron replacement therapy was initiated between August 2016 and June 2023 and continued for at least seven months. All patients underwent hemodialysis for 4–5 h three times a week.

All patients provided written informed consent permitting data sampling and analysis. The protocol for the study was approved by the ethics committee of Biomarker Society, Inc., which comprises five committee members, including outside experts.

### 4.2. Methods

Patients with serum ferritin <60 ng/mL and Hb <12 g/dL received iron replacement therapy as 50–120 mg of iron daily in the form of ferrous citrate (50 or 100 mg/day) or ferric citrate (250 or 500 mg/day; 60 or 120 mg of iron).

All blood samples were collected at the start of hemodialysis at the beginning of the week. Blood examinations were performed twice a month, and patients were provided DA, long-acting ESAs to maintain Hb at the target level of 10–12 g/dL, in accordance with the guidelines of the Japanese Society for Dialysis Therapy. Iron-related parameters were determined based on blood samples collected once a month, with TSAT calculated from serum iron (Fe) and total iron-binding capacity (TIBC), using the following equation. TSAT = Fe/TIBC × 100

TBI was calculated as the sum of RBC iron and iron stores [17]. BSA was calculated using the Du Bois formula [44]. Patients were regarded as iron-therapy responders (i.e., patients for whom iron replacement therapy was effective) when (1) their Hb level was elevated by at least 1 g/dL, (2) their Hb level was at least 12 g/dL, and/or (3) the DA dose to maintain Hb at the target level could be reduced [15]. TBI and erythrocyte/iron-related parameters were evaluated in iron-therapy responders and non-responders on a monthly basis through month 7. TBI was regarded as achieving a steady state during the course of therapy when the trend shifted from an increase and the difference from the value at one month was less than or equal to zero; however, in cases where Hb at a steady-state was >12 g/dL, the subsequent TBI when Hb was ≤12 g/dL was regarded as the steady-state value. We then evaluated the correlations and the erythrocyte/iron-related parameters at the steady-state TBI timepoint.

To investigate the correlation between hepcidin and TBI, we also targeted 148 HD patients who visited our institution in September 2008 and underwent hepcidin, Hb, and serum ferritin measurements and had not received intravenous iron in the previous three months, for evaluation in a sensitivity analysis based on their blood sample data. The sensitivity analysis was performed using the same method as in a previous study [25], and the hepcidin cut-off values were set with 5 ng/mL intervals between 10 and 30 ng/mL. Hepcidin was measured using a quantitative method involving liquid chromatography coupled with tandem mass spectrometry [45].

TBI is a novel indicator for estimating in vivo iron levels, proposed by Cable et al. Most of the body’s iron is accounted for by RBC iron (the iron contained in RBC Hb) and iron stores, so the sum of these parameters is considered to represent the approximate amount of iron in vivo. The amount of iron in 1 g of Hb is assumed to be 3.4 mg and the estimated blood volume (EBV) was calculated using Nadler’s formula, which utilizes height, weight, and sex, as follows [17].

TBI (mg) = RBC iron + iron stores

RBC iron (mg) = 3.4 mg × Hb value (g/dL) × 10 × EBV (L) × 0.91

Iron stores (mg) = (−13.8588 + 0.3929 × G + 15.5999log_10_(F) − 2.0519(log_10_(F))^2^) × body weight (kg)

F: ferritin (ng/mL); G: men = 0, women = 1.

### 4.3. Statistical Analysis

All statistical analyses were performed using SAS ver. 9.4 (SAS Institute, Cary, NC, USA). Data are presented as the mean ± SD and the median with interquartile range. The *t*-test was used to compare groups in terms of normally distributed continuous variables, and the Mann–Whitney U test was used for other skewed continuous variables. The chi-square test was used to compare nominally scaled variables. One-way repeated measures analysis of variance was performed to compare changes over time, and Bonferroni’s multiple comparison test was used for post hoc tests. Two-way repeated measures analysis of variance was performed to compare changes over time between iron-therapy responders and non-responders. Pearson’s product-moment correlation coefficient and a generalized linear regression model were used. The differences among the slopes in the regression models were also tested using the generalized linear regression model with the interaction term. To clarify the relationship between TBI and hepcidin, both linear and nonlinear regression models were applied. To determine the optimal cut-off values, we performed a sensitivity analysis using Pearson’s correlation coefficient and a generalized linear regression model involving the interaction term. The F statistic is a measure of the difference in variance between two groups. A tool for comparing how far individual data from different groups deviate from the average. In this paper, we apply the F statistic to use the value where the slope changes the most as the optimal cut point. Two-tailed *p*-values less than 0.05 were considered to indicate statistically significant differences.

## 5. Conclusions

This study showed that hemodialysis patients can increase their body iron levels during oral iron replacement therapy and maintain them at the same level after sufficiency has been achieved. The results suggest that hepcidin may be a marker indicative of TBI sufficiency, but we consider that the use of serum ferritin as the sole marker indicative of iron sufficiency is hindered by the fluctuation of Hb in HD patients. However, regression equations using BSA showed a high degree of fit. Therefore, BSA might potentially be used in estimations of TBI sufficiency, which may be useful for avoiding excessive iron dosage.

## Figures and Tables

**Figure 1 ijms-25-01508-f001:**
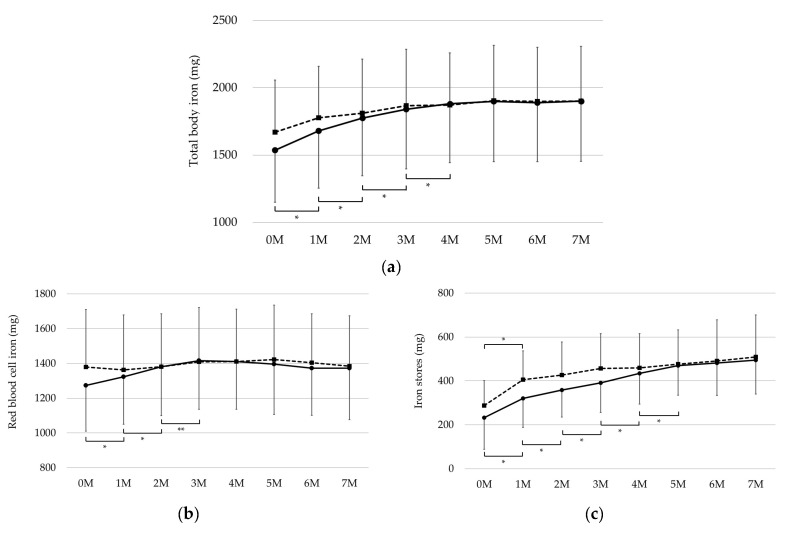
Changes over time in (**a**) TBI, (**b**) RBC iron, (**c**) iron stores, (**d**) serum ferritin, and (**e**) DA/BW; Darbepoetin α per body weight. 
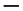
: iron-therapy responders, 

: non-responders, *: *p* < 0.01, **: *p* < 0.05.

**Figure 2 ijms-25-01508-f002:**
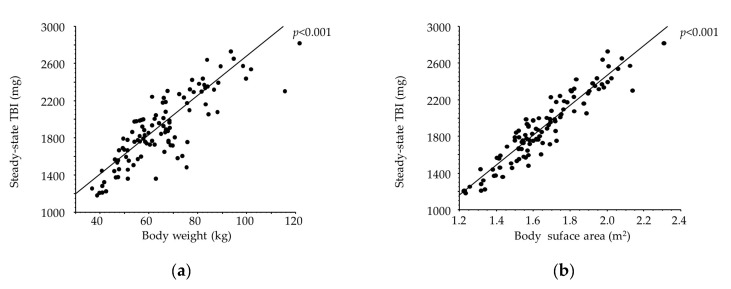
Simple linear regression analysis for steady-state total body iron (TBI). (**a**) Body weight (BW, kg), TBI = 563.769 + 21.187 × BW; R^2^ = 0.771, (**b**) body surface area (BSA, m^2^), TBI = −791.914 + 1628.606 × BSA; R^2^ = 0.883, (**c**) hemoglobin, and (**d**) serum ferritin.

**Figure 3 ijms-25-01508-f003:**
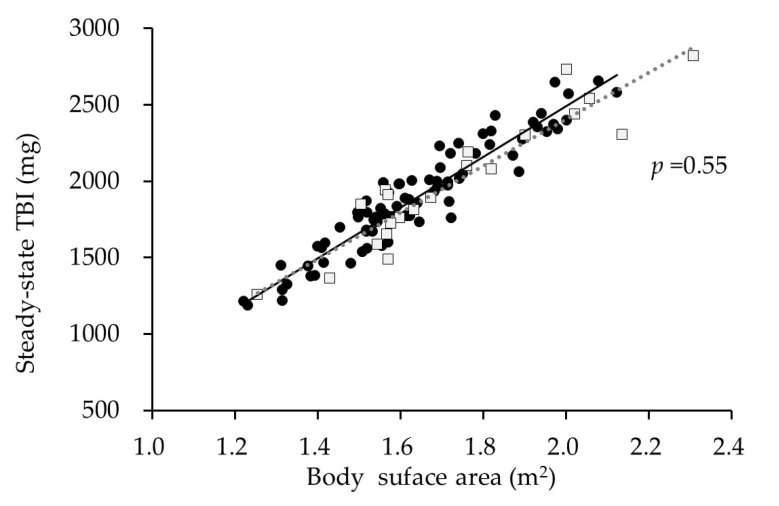
Comparison of the correlation with steady-state TBI and BSA between iron-therapy responders and non-responders. 
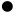
, 
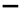
: iron-therapy responders, 
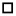
, 
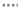
: non-responders.

**Figure 4 ijms-25-01508-f004:**
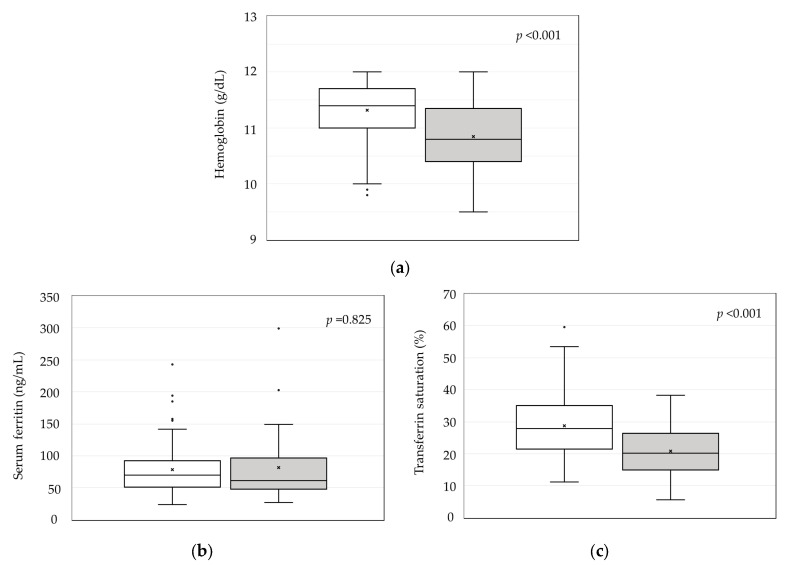
Comparison of (**a**) hemoglobin, (**b**) serum ferritin, and (**c**) transferrin saturation between iron-therapy responders and non-responders at steady-state. 
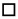
: responders, 
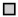
: non-responders, ×: mean value, 
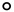
: outlier.

**Figure 5 ijms-25-01508-f005:**
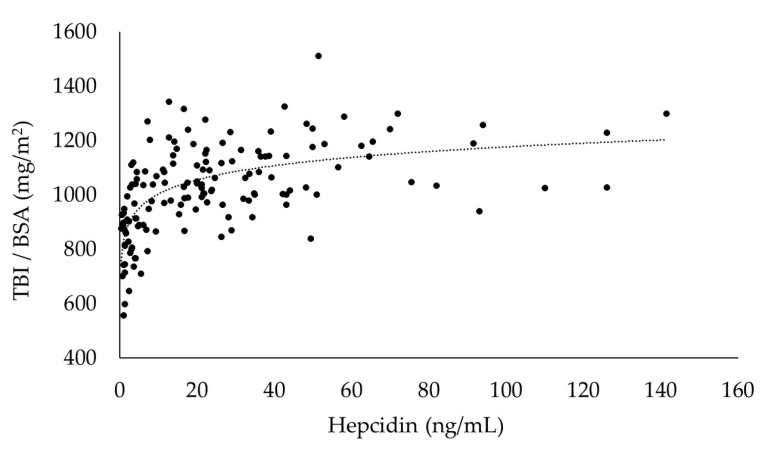
Correlation between BSA-adjusted TBI and hepcidin. y = 75.234ln(x) + 830.77, R^2^ = 0.400.

**Table 1 ijms-25-01508-t001:** Patient characteristics.

Variables	All	Responders	Non-Responders	*p*-Value
Age (years)	68.1 ± 12.6	68.5 ± 12.7	67.1 ± 12.2	0.63
Gender				
Men	75	58	17	0.80
Women	30	22	8	
IRT courses	105	80	25	
Duration of vintage (years) †	6.6 (2.4–17.4)	7.6 (2.1–17.1)	5.2 (3.0–18.2)	0.96
Body surface area (m^2^)	1.69 ± 0.41	1.68 ± 0.45	1.71 ± 0.24	0.76
Primary diagnosis				
Chronic glomerulonephritis	32	24	8	0.08
Diabetes nephropathy	35	25	10	
Renal sclerosis	20	13	7	
Polycystic kidney disease	3	3	0	
Other	15	15	0	
Hb (g/dL)	10.4 ± 0.7	10.4 ± 0.7	10.5 ± 0.7	0.45
RBC (×10^4^/μL)	344.3 ± 36.5	344.0 ± 38.0	345.4 ± 31.6	0.87
MCH (pg)	30.3 ± 2.1	30.3 ± 2.3	30.4 ± 1.4	0.79
MCV (fL)	96.1 ± 5.5	96.1 ± 5.9	95.9 ± 4.2	0.84
s-Fe (μg/dL)	50.9 ± 17.2	50.4 ± 17.9	52.6 ± 14.8	0.59
TIBC (μg/dL)	284.2 ± 40.5	282.3 ± 40.5	290.5 ± 40.8	0.38
TSAT (%)	18.1 ± 5.9	18.1 ± 6.2	18.3 ± 5.0	0.85
Serum ferritin (ng/mL)	27.3 ± 12.5	26.7 ± 13.0	29.3 ± 10.3	0.36
Albumin (g/dL)	3.5 ± 0.3	3.4 ± 0.2	3.5 ± 0.3	<0.05
C-reactive protein (mg/dL) †	0.10 (0.04–0.24)	0.09 (0.04–0.17)	0.19 (0.09–0.47)	<0.05
nPCR	0.89 ± 0.16	0.87 ± 0.16	0.93 ± 0.19	0.44
Kt/V	1.49 ± 0.26	1.49 ± 0.26	1.47 ± 0.26	0.80
Darbepoetin α (μg/week)	20 (10–30)	20 (15–40)	10 (10–20)	<0.01
Oral iron preparation				
Sodium ferrous citrate	64	51	13	0.35
Ferric citrate hydrate	41	29	12	

Values are shown as the number, mean ± standard deviation, or median (interquartile range) †. IRT; iron replacement therapy, Hb; hemoglobin, RBC; red blood cells, MCH; mean corpuscular hemoglobin, MCV; mean corpuscular volume, TIBC; total iron-binding capacity, TSAT; transferrin saturation, nPCR; normalized protein catabolic rate, Kt/V; urea removal status indicator.

**Table 2 ijms-25-01508-t002:** Comparison of TBI, RBC iron, and iron stores at baseline between iron-therapy responders and non-responders.

Variables	Responders	Non-Responders	*p*-Value
Total body iron (mg)	1535.7 ± 389.8	1669.5 ± 393.8	0.15
Red blood cell iron (mg)	1274.2 ± 263.2	1379.8 ± 330.7	0.10
Stored iron (mg)	232.3 ± 143.8	288.1 ±113.2	0.08

**Table 3 ijms-25-01508-t003:** Sensitivity analysis for hepcidin level with BSA-adjusted TBI.

Optimal Cut Point	Correlation Coefficient (r)	(95% Confidence Interval)	*p*-Value	F-Statistic for Interaction
≤10	25.79	(11.54 to 40.03)	<0.001	12.58
>10	1.05	(0.12 to 1.97)	0.027	
≤15	25.01	(17.60 to 32.55)	<0.001	38.74 *
>15	1.33	(0.34 to 2.32)	0.009	
≤20	15.65	(10.65 to 20.64)	<0.001	31.87
>20	1.27	(0.21 to 2.33)	0.020	
≤25	11.38	(8.05 to 14.71)	<0.001	29.04
>25	1.25	(−0.09 to 2.59)	0.066	

*: optimal cut-off point.

## Data Availability

All data are contained within the article.

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
