# Peer review of "High Ferritin Is Not Needed in Hemodialysis Patients: A Retrospective Study of Total Body Iron and Oral Iron Replacement Therapy"

_ijms, 2024, doi:10.3390/ijms25031508_

Round 1

Reviewer 1 Report

Comments and Suggestions for Authors

Author Response

Review IJMS-2806485 

The authors address a very interesting topic. High ferritin values do not represent a marker of physical fitness, but are an inflammatory biomarker. The DOPPS (rightly cited by the authors) and all post-hoc analyzes have highlighted a correlation between high ferritin levels and mortality outcomes in hemodialysis patients.

But the DOPPS study is old and many references are old. The authors referred to old and contradictory literature. In reality, the PIVOTAL study stated the opposite of DOPPS. But it is not mentioned. In this study, published in two phases (with revision of the statistics), administering iron to hemodialysis patients meant improving survival outcomes.

The paper needs to be improved.

A paragraph on the mechanism of ferroportin and the inflammatory state must be added, the classification of iron-deficiency anemias is missing and there is no reference to ferric carboxymaltose. The literature in the last three years has produced many results on ferric carboxymaltose and the improvement of the inflammatory state of the hemodialysis patient. –

Lacquaniti A, Gargano R, Campo S, Casuscelli di Tocco T, Schifilliti S, Monardo P. The Switch from Ferric Gluconate to Ferric Carboxymaltose in Hemodialysis Patients Acts on Iron Metabolism, Erythropoietin, and Costs: A Retrospective Analysis. Medicina (Kaunas). 2023 Jun 2;59(6):1071. doi: 10.3390/medicina59061071. PMID: 37374275; PMCID: PMC10305135. –

Lacquaniti A, Pasqualetti P, Tocco TCD, Campo S, Rovito S, Bucca M, Ragusa A, Monardo P. Ferric carboxymaltose versus ferric gluconate in hemodialysis patients: Reduction of erythropoietin dose in 4 years of follow-up. Kidney Res Clin Pract. 2020 Sep 30;39(3):334-343. doi: 10.23876/j.krcp.20.015. PMID: 32839355; PMCID: PMC7530360.

Thank you for your suggestions.

We added the sentences as follows in “Introduction”. (P2, line 49-53) (P2, line 54-56)

Iron supply to the blood is mainly provided by intestinal cells, reticuloendothelial macrophages, and hepatocytes via FPN. When hepcidin binds to ferroportin, the conjugate is degraded in lysosomes and iron cannot be transported from the cell to the bloodstream.

Iron deficiency is classified into absolute iron deficiency, in which the total body iron is deficient, and functional iron deficiency, in which iron deficiency in the blood occurs due to elevated hepcidin despite sufficient total body iron.

We added the sentence as follows in “Discussion” and reference 33-43. (P9, line 266-282), (P13, line 465-497)

Recently, the PIVOTAL study reported that high-dose intravenous iron administration is effective in improving prognosis and reducing the dose of ESA [33].   In addition, intravenous formulations of high-titer iron are now available on the market, allowing large amounts of iron to be administered intravenously in a single dose. There are also many reports of the effects of Ferric carboxymaltose and ferric derisomaltose, which can be administered intravenously in large doses by reducing oxidative stress, on anemia improvement, ESA reduction, and increase in serum ferritin and TSAT [34-40]. However, it has been suggested that high-dose iron administration and high ferritin level are associated with iron deposition in the liver [41, 42] and iron deposition in the liver and heart has also been reported to be observed in many hemodialysis patients [43]. Therefore, although high-dose intravenous iron administration may be useful in the short term, concerns remain about its long-term safety. In this study, the average serum ferritin and TSAT in the steady state of responders group were 78.6 ng/mL and 28.7%, respectively, and high TSAT was maintained even with low serum ferritin, indicating good iron utilization efficiency. It has been shown that in such patients, it is possible to improve anemia and reduce ESA dosage even with low-doses of oral iron supplements and low ferritin level.

Furthermore, the authors do not refer to erythropoietin requirements and biomarkers of inflammatory status (including ferritin). In short, the paper must be modified, starting from the search for literature that is too old and does not include new references. The discussion must be broadened and a paragraph on the need for ESA in hemodialysis patients should be introduced. Without these changes the paper cannot be published

Thank you for your suggestions.

We added the changes in ferritin and dose of darbopoetin α per body weight on Fig 1 and the sentence as follows in “Results”. (P4, line 135-140)

On the other hand, serum ferritin showed a significant increase in both groups at month 1, but thereafter a significant increase was only observed at month 5 in the responder group, and the change over time was not different between the two groups (p = 0.72, Figure 1d). DA per body weight showed a significant decrease at month 3 and 4 in the effective group, but a slight increase in the non-responder group, and the change over time was also significantly different between the two groups (p<0.001, Figure 1e).

We added a sentence in “Introduction”, a paragraph in “Discussion” on the need for ESA in hemodialysis patients as follows and references 21-24. (P2, line 57-60), (P8, line 179-197), (P12, line 433- P13, 443)

Renal anemia in HD patients results from decreased EPO production due to deficiency of HIF2α. Therefore, therapeutic approaches to renal anemia in hemodialysis (HD) patients center on HIF–prolyl hydroxylase domain inhibitors, which have recently become commercially available, as well as erythropoietin-stimulating agents (ESAs).

Renal anemia is an almost inevitable complication of CKD patients and is defined as anemia resulting from impaired production of erythropoietin (EPO) in the kidney; a deficiency of EPO reduces hematopoietic stimulation of the bone marrow, resulting in anemia. As for molecular biological mechanisms, the EPO protein was purified, EPO-producing cells in the kidney were identified in 2008 [21], and a series of studies on hypoxia-inducible factors revealed that reduced production of EPO is induced by weakened HIF2α [22, 23]. On the other hand, in the past, its treatment required frequent blood transfusions, leading to iron accumulation and the risk of infectious diseases such as hepatitis, which particularly impaired the quality of life of dialysis patients. However, in 1990, the clinical application of synthetic erythropoietin (recombinant human erythropoietin: rHuEPO), now collectively called ESA (erythropoietin stimulating agent), became possible, especially the clinical application of long-acting ESA has had a therapeutic effect. Furthermore, since 2019, HIF-PH inhibitors have been available for clinical use [24]. Thus, the treatment of renal anemia, along with the application of various preparations, now involves the establishment of optimal Hb levels and, in particular, information on iron supply, which is essential for efficient hematopoiesis. Progress has also been made in identifying the regulatory mechanisms of iron metabolism and related proteins, and understanding iron kinetics and adequate iron supply, along with ESA or HIF-PHI administration, are essential prerequisites for stable renal anemia management.

Reviewer 2 Report

Comments and Suggestions for Authors

The current therapy of renal anemia targets also the iron deficiency, and even if many progresses have been made in developing new forms of drugs to better correct Hb levels, the optimal iron level of HD patients is unclear, considering that serum ferritin (the most used marker to determine the grade of this deficiency) is affected by chronic inflammation, as you clearly highlighted. Furthermore, as you stated, RBC iron and iron stores function together as the two major in vivo reservoirs of iron, and serum ferritin does not necessarily reflect body iron levels in individuals with unstable hematopoiesis, as frequently can be noticed in chronic dialyzed population. Therefore, the present work could improve our current knowledge. The design of the study is properly described and the conclusions are supported by your findings, which suggest the importance of adding supplementary variables (such as BSA) to determine TBI sufficiency for avoiding excessive iron dosage. I congratulate you for a very well conducted research, but there are some few recommendations that maybe will improve your article:

- The first row of Table 1 is confusing; usually N refers to the total number of subjects. For a better understanding, please redefine the first row as IRT (iron replacement therapy) courses and maybe it should be placed after age and gender (better use this term instead of sex). Perhaps “duration of dialysis” should be defined as “dialysis vintage”. When using “m2”, please use the superscript (m2). Considering the manner of the variables presentation, the term “polycystic” should start with uppercase letter (Polycystic); the same for “serum ferritin”.

- When presenting the methodology, you mentioned that the blood samples were collected at the beginning of the relevant week – please explain this term (“relevant week”); why a specific week was selected.

- As you considered important to present the value of F-statistic for interaction, please explain this term in the statistical analysis – primarily, your article targets the medical community, therefore statistical terms should be carefully presented for a better understanding of the results.

Author Response

The current therapy of renal anemia targets also the iron deficiency, and even if many progresses have been made in developing new forms of drugs to better correct Hb levels, the optimal iron level of HD patients is unclear, considering that serum ferritin (the most used marker to determine the grade of this deficiency) is affected by chronic inflammation, as you clearly highlighted. Furthermore, as you stated, RBC iron and iron stores function together as the two major in vivo reservoirs of iron, and serum ferritin does not necessarily reflect body iron levels in individuals with unstable hematopoiesis, as frequently can be noticed in chronic dialyzed population. Therefore, the present work could improve our current knowledge. The design of the study is properly described and the conclusions are supported by your findings, which suggest the importance of adding supplementary variables (such as BSA) to determine TBI sufficiency for avoiding excessive iron dosage. I congratulate you for a very well conducted research, but there are some few recommendations that maybe will improve your article:

- The first row of Table 1 is confusing; usually N refers to the total number of subjects. For a better understanding, please redefine the first row as IRT (iron replacement therapy) courses and maybe it should be placed after age and gender (better use this term instead of sex). Perhaps “duration of dialysis” should be defined as “dialysis vintage”. When using “m2”, please use the superscript (m2). Considering the manner of the variables presentation, the term “polycystic” should start with uppercase letter (Polycystic); the same for “serum ferritin”.

Thank you for your suggestions.

We rewrote the points pointed out in Table 1.  (P2-3, Table 1)

- When presenting the methodology, you mentioned that the blood samples were collected at the beginning of the relevant week – please explain this term (“relevant week”); why a specific week was selected.

Thank you for your suggestions.

We removed “relevant”. (P10, line 304-305)

- As you considered important to present the value of F-statistic for interaction, please explain this term in the statistical analysis – primarily, your article targets the medical community, therefore statistical terms should be carefully presented for a better understanding of the results.

Thank you for your suggestions.

We added the sentences as follows in “statistical analysis”. (P11, line 357-360)

The F statistic is a measure of the difference in variance between two groups. A tool for comparing how far individual data from different groups deviate from the average. In this paper, we apply the F statistic to use the value where the slope changes the most as the optimal cut point.

Reviewer 3 Report

Comments and Suggestions for Authors

The paper presents data showing sufficiency of iron stores for effective erythropoiesis in HD patients with lower than 60 ng/mL levels of serum ferritin.

This is probably true, but the data analysis does not show an important measures of effectiveness of darbopoetin therapy - the dose per kg of body mass and correlation between the measure and ferritin level. 

Moreover, oral iron supplementation is poorly tolerated by patients. As the consequence the adherence to the therapy is poor. It would be nice to show the data concerning adherence /non-adherence and ADRs in the analysed cohort.

Please modify discussion concerning i.v. and oral iron therapy concerning tolerance and adherence. 

Could you present changes in ferritin and dose of darbopoetin per kg with time as for other parameters shown on Fig 1.

Data from Figure 4 could be presented in Table 2. Low number of non-respondents (25) decreasing importance of not-significant finding. 

Some presented correlations: BSA-adjusted TBI and hepcidin - what is the reason for BSA-adjustment?

Author Response

The paper presents data showing sufficiency of iron stores for effective erythropoiesis in HD patients with lower than 60 ng/mL levels of serum ferritin.

This is probably true, but the data analysis does not show an important measures of effectiveness of darbopoetin therapy - the dose per kg of body mass and correlation between the measure and ferritin level. 

Moreover, oral iron supplementation is poorly tolerated by patients. As the consequence the adherence to the therapy is poor. It would be nice to show the data concerning adherence /non-adherence and ADRs in the analysed cohort.

Please modify discussion concerning i.v. and oral iron therapy concerning tolerance and adherence. 

Thank you for your suggestion.

We added the sentence as follows in “Results”. (P3, line 101-106)

During the observation period, iron supplementation was administered intravenously to 17 patients due to gastrointestinal disorders, and 124 patients received oral iron supplementation.    

The target population consisted of 88 HD patients who had undergone a total of 111 courses of iron therapy over seven months, excluding patients with hemorrhage and increased ferric citrate hydrate dose due to high phosphorus levels.

Could you present changes in ferritin and dose of darbopoetin per kg with time as for other parameters shown on Fig 1.

Thank you for your suggestion.

We have added the changes in ferritin and dose of darbopoetin per body weight on Fig 1 and the sentence as follows in “Results”. (P4, line 135-140)

On the other hand, serum ferritin showed a significant increase in both groups at month 1, but thereafter a significant increase was only observed at month 5 in the responder group, and the change over time was not different between the two groups (p = 0.72, Figure 1d). DA per body weight showed a significant decrease at month 3 and 4 in the effective group, but a slight increase in the non-responder group, and the change over time was also significantly different between the two groups (p<0.001, Figure 1e).

Data from Figure 4 could be presented in Table 2. Low number of non-respondents (25) decreasing importance of not-significant finding. 

Thank you for your suggestion.

Sorry for the confusion.

Table 2 is the data at baseline, and Figure 4 is the data at steady-state. We added “at baseline” in the title of Table 2 and "at steady-state " in the legend of Figure 4. (P5, line 145), (P7, line 167)

BSA data was duplicated in Tables 1 and 2. BSA data in Table2 has been deleted.

Some presented correlations: BSA-adjusted TBI and hepcidin - what is the reason for BSA-adjustment?

Thank you for your suggestion.

We have added the reason for adjusting TBI with BSA in “Discussion” (P9, line 253-256).

This study showed that the required TBI strongly dependent on BSA. Therefore, the BSA-adjusted TBI is expected to represent the degree of TBI sufficiency. If TBI is sufficient, TBI is presumed to remain constant with hepcidin-induced mucosal block.

Round 2

Reviewer 1 Report

Comments and Suggestions for Authors

The authors improved the manuscript. The paper can be published

Reviewer 3 Report

Comments and Suggestions for Authors

The paper was improved alongside the comments.

No further comments.